# Absent Metabolic Transition from the Early to the Late Period in Non-Survivors Post Cardiac Surgery

**DOI:** 10.3390/nu14163366

**Published:** 2022-08-17

**Authors:** Cecilia Veraar, Arabella Fischer, Martin H. Bernardi, Isabella Sulz, Mohamed Mouhieddine, Martin Dworschak, Edda Tschernko, Andrea Lassnigg, Michael Hiesmayr

**Affiliations:** 1Department of Anesthesiology, Intensive Care Medicine and Pain Medicine, Division of Cardiac Thoracic Vascular Anesthesia and Intensive Care Medicine, Medical University of Vienna, 1090 Vienna, Austria; 2Center for Medical Statistics, Institute for Medical Statistics, Informatics and Intelligent Systems, Medical University Vienna, 1090 Vienna, Austria

**Keywords:** resting energy expenditure, oxygen consumption, oxygen delivery, cardiac surgery, intensive care medicine, mitochondrial dysfunction

## Abstract

After major surgery, longitudinal changes in resting energy expenditure (REE) as well as imbalances in oxygen delivery (DO_2_) and distribution and processing (VO_2_) may occur due to dynamic metabolic requirements, an impaired macro- and microcirculatory flow and mitochondrial dysfunction. However, the longitudinal pattern of these parameters in critically ill patients who die during hospitalization remains unknown. Therefore, we analyzed in 566 patients who received a pulmonary artery catheter (PAC) their REE, DO_2_, VO_2_ and oxygen extraction ratio (O_2_ER) continuously in survivors and non-survivors over the first 7 days post cardiac surgery, calculated the percent increase in the measured compared with the calculated REE and investigated the impact of a reduced REE on 30-day, 1-year and 6-year mortality in a uni- and multivariate model. Only in survivors was there a statistically significant transition from a negative to a positive energy balance from day 0 until day 1 (Day 0: −3% (−18, 14) to day 1: 5% (−9, 21); *p* < 0.001). Furthermore, non-survivors had significantly decreased DO_2_ during the first 4 days and reduced O_2_ER from day 2 until day 6. Additionally, a lower REE was significantly associated with a worse survival at 30 days, 1 year and 6 years (*p* = 0.009, *p* < 0.0001 and *p* = 0.012, respectively). Non-survivors seemed to be unable to metabolically adapt from the early (previously called the ‘ebb’) phase to the later ‘flow’ phase. DO_2_ reduction was more pronounced during the first three days whereas O_2_ER was markedly lower during the following four days, suggesting a switch from a predominantly limited oxygen supply to prolonged mitochondrial dysfunction. The association between a reduced REE and mortality further emphasizes the importance of REE monitoring.

## 1. Introduction

Energy in the form of adenosine triphosphate [1] and other high-energy compounds are required for all cellular activities; it is most efficiently obtained via the oxidation of nutrients. Macro- and microcirculation as well as mitochondrial functioning are key components to maintain a normal cellular physiology [2]. Therefore, oxygen delivery (DO_2_) as a measure of macrocirculation and oxygen consumption (VO_2_) as a combined measure of the microcirculatory distribution of the blood flow and mitochondrial activity are essential to evaluate the metabolic state [3].

A critical fall in VO_2_ and, therefore, resting energy expenditure (REE) induces reversible impairments and, finally, irreversible alterations that result in cell death. Maintaining sufficient oxygen availability for the cell is, therefore, fundamental for cell survival [4]. In septic and cardiac arrest patients, decreased VO_2_ was associated with a higher mortality rate [5,6]. This association was initially thought to be mainly related to inadequate DO_2_ and is now explained by an impaired oxygen extraction ratio (O_2_ER) due to an altered mitochondrial function [7,8].

Metabolic, endocrine and immunological reactions are triggered by surgical trauma and extracorporeal circulation (ECC) [9]. The metabolic response to trauma over time was first described by Sir David Cuthbertson who coined the terms ‘ebb’ and ‘flow’ phases, describing an initially reduced metabolic rate followed by a later increase after an injury [10]. A third chronic phase has more recently been proposed [11]. In the latest update of the ESPEN guidelines in 2019, the metabolic response to trauma was redefined. The ‘ebb’ phase was renamed the early period and defined by catabolism, a lower body temperature and reduced VO_2_, aiming to reduce post-traumatic energy depletion. The former ‘flow’ phase is now described as the late period and is followed by the chronic phase, which devolves from a catabolic to an anabolic status. The transition from one state to the next depends on the injury severity and is associated with stress, muscle atrophy, the administration of further medication (catecholamine, sedatives, neuromuscular blocking agents), mechanical ventilation and renal replacement therapy [12].

During the early post-injury period, the REE is usually lower than before the injury. In contrast, in the later phase, the REE may even increase to values higher than before the injury [13,14].

During these transitions from one period to another, further factors impact the REE, including physiological derangements such as fever, hypothermia, changes in the heart rate, shivering, agitation, infections and fasting status as well as therapeutic interventions such as catecholamine support, sedatives, non-selective β-blockers and active cooling [15]. Therefore, it remains difficult to estimate the net balance of the REE in critically ill patients and the metabolic patterns of patients with adverse outcomes remain largely unknown.

Thus, in this cohort study, we evaluated whether hyper- or hypometabolic states predominate in survivors and non-survivors. We analyzed the relation between the measured and calculated REE over time. Furthermore, we investigated the longitudinal dynamics of DO_2_, VO_2_ and O_2_ER, measured via a pulmonary artery catheter (PAC) after different cardiac procedures over the first 7 days. Finally, we investigated in a uni- and multivariable model the association of the REE with mortality at 30 days, 1 year and 6 years after cardiac surgery.

## 2. Methods and Materials

### 2.1. Ethical Approval

The study was conducted in accordance with the Declaration of Helsinki (as revised in 2013) and was approved by the Ethics Committee of the Medical University of Vienna (EK1099/2022). The data collection was performed in accordance with the approved ethical guidelines.

### 2.2. Study Design and Patients

This work was designed as a single-center cohort study. We enrolled 566 consecutive patients from 2012–2015 who underwent elective or emergency heart surgery with a PAC for hemodynamic monitoring. The data on survival time were determined in April 2022. The longest follow-up time, either observed or censored, was 6 years.

A decision on PAC insertion was based on institutional practice and individual physician risk evaluations. We included all PAC measurements performed within the first 7 days after surgery and excluded all patients younger than 18 years, patients requiring extracorporeal membrane oxygenation and right heart assist devices. The PACs were inserted using the Seldinger technique, usually through a right internal jugular approach. Correct positioning with the proximal port located in the SVC and the distal port in the PA was confirmed via an X-ray. The PAC measurements were recorded every 10 min. In-hospital mortality was used to divide the patients into survivors and non-survivors. The calculations are depicted in Table 1.

### 2.3. Statistical Analysis

The demographic data were presented using descriptive statistics. The mean ± standard deviation (SD) were given for the continuous variables. The categorical variables were shown as a frequency (percentage). Variables such as CO and SvO_2_ were determined using the PAC and VO_2_, REE, O_2_ER and REE were calculated as shown in Table 1.

A Student’s *t*-test was applied for the parametric data, the Mann–Whitney U test was used for unpaired non-normally distributed data and repeated measure ANOVA testing was performed for the multiple comparison analysis.

All variables comprising CO, SvO_2_, VO_2_, REE, O_2_ER and DO_2_ were averaged for each of the first 7 days and presented as a median and interquartile range (25% percentile and 75% percentile, respectively) for in-hospital survivors and non-survivors. In addition, we calculated the percent increase in the measured REE via the PAC compared with the predicted REE calculated via the formula for survivors and non-survivors, respectively. Furthermore, the REE was averaged over the first 7 days. Patients were divided into two groups depending on their REE being either above or below the median REE of 1640 kcal/d. A survival analysis was performed using the Kaplan–Meier analysis and logrank test. Uni- and multivariate Cox regression analyses were calculated for the 30-day, 1-year and 6-year mortality. The data were shown as a hazard ratio (HR) and 95% confidence interval. All tests were two-sided and *p*-values < 0.05 were considered to be statistically significant.

The statistical analyses were performed using R 3.3.1 and SPSS (version 28.0; IBM SPSS Inc., Chicago, IL, USA). The figures were plotted using GraphPad Prism (version 8.0; GraphPad Software Inc., San Diego, CA, USA).

### 2.4. Data Availability

All data generated or analyzed during this study are included in the published article and its Appendix A.

## 3. Results

In this retrospective cohort study, we analyzed 566 ICU patients over the first 7 days post cardiac surgery. The demographic and clinical data are shown in Table 2. Of the participants, 27% of all patients underwent single-valve procedures, 12% received CABG surgery, 17% underwent CABG and a valve procedure, 3% obtained a vascular graft, 16% were LVAD and 19% were HTX patients; 5% received other procedures.

### 3.1. Absent Metabolic Transition in Non-Survivors from the Early ‘Ebb Phase to the Late ‘Flow’ Phase

In survivors, compared with non-survivors, the measured energy balance significantly increased from day 0 until day 4, but not on days 5, 6 and 7, as shown in Figure 1A. In survivors, we found a negative measured energy balance of −3% compared with the predicted REE on day 0, which subsequently rose significantly until day 4 (Day 0: −3% (−18, 14) compared with day 1: 5% (−9, 21), day 2: 4% (−10, 21), day 3: 4% (−10, 22) and day 4: 5% (−10, 23); *p* < 0.001), as demonstrated in Figure 1B. In non-survivors, there was a negative measured energy balance compared with the predicted REE from day 0 until day 7 (Day 0: −17% (−55, 5), day 1: −23% (−42, −3), day 2: −20% (−43, −2), day 3: −16% (−33, −2), day 4: −16% (−41, 3), day 5: −11% (−32, 17), day 6: −19% (−38, 6) and day 7: −20% (−39, 10)), as shown in Figure 1C.

### 3.2. Increased SvO_2_ and Reduced CCO, VO_2_, REE, O_2_ER and DO_2_ Levels in Non-Survivors Compared with Survivors over the First Seven Days

The REE was significantly reduced on days 0, 1, 2, 3 and 4, but not on days 5, 6 and 7 post cardiac surgery in non-survivors relative to survivors, as depicted in Figure 2A.

Non-survivors had solely significantly lower O_2_ER values on post-operative days 2, 3, 4 and 6 compared with survivors after heart surgery, as demonstrated in Figure 2B.

DO_2_ was significantly reduced in non-survivors compared with survivors during the first four post-operative days, as shown in Figure 2C.

VO_2_ significantly decreased within the first four days after surgery in non-survivors compared with survivors, as pictured in Figure 2D.

CCO was significantly lower in non-survivors in contrast to survivors within the first five post-operative days, as illustrated in Figure 2E.

There was no difference in SvO_2_ on day 0 after surgery, but from day 1 until day 7 SvO_2_ significantly rose in non-survivors compared with survivors, as depicted in Figure 2F.

Details on the statistically significant differences in the REE, O_2_ER, DO_2_, VO_2_, CCO and SvO_2_ between survivors and non-survivors are shown in Appendix A.

### 3.3. Increased 30-Day, 1-Year and 6-Year Mortality in Patients with a Reduced REE

In the Kaplan–Meier survival analysis for 30 days, 1 year and 6 years we found a significantly reduced overall survival rate for patients with a reduced REE, which was below the median of 1640 kcal/day (*p* = 0.009, *p* < 0.0001 and *p* = 0.012, respectively; Figure 3). Furthermore, in the non-parametric group testing, a low REE was not associated with a low BMI, size and body weight (*p* = 0.190, *p* = 0.374 and *p* = 0.104, respectively).

### 3.4. Univariate and Multivariate Cox Regression Analyses for 30 Days, 1 Year and 6 Years after Cardiac Surgery

A REE ≤ 1640 kcal/d was associated with increased mortality in the univariate model and remained an independent factor in the multivariate analysis for 30 days, 1 year and 6 years after cardiac surgery. Age did not impact 30-day mortality, but was associated with increased mortality at 1 year and 6 years after surgery in the uni- and multivariate analyses. The BMI did not influence the outcome 30 days and 1 year post-surgery; however, after 6 years a BMI between 25 and 30 was associated with significantly decreased mortality compared with patients with a BMI < 25 in the univariate, but not in the multivariate, cox regression analysis. An increased ECC time (>170 min) was associated with a higher mortality rate after 30 days, 1 year and 6 years post-surgery in the uni- and multivariate cox regression analyses. One and six years following surgery, patients with a minimum Hb < 8 mg/dL had a significantly increased mortality in the univariate, but not in the multivariate, analysis. In the univariate analysis for 1-year mortality, the hazard ratio of maximum lactate levels > 3.6 mmol/L was significantly higher. Furthermore, patients receiving > 3 PRBCs had a significantly increased hazard ratio for 1-year and 6-year mortality in the uni- and multivariate analyses (Table 3).

## 4. Discussion

In this cohort study, we observed the following. (1) There was an inadequate metabolic response to stress in patients with early adverse outcomes. The metabolic response of non-survivors failed to transition from the early previously-called ‘ebb’ phase to the late previously-called ‘flow’ phase. (2) DO_2_ and O_2_ER levels were decreased in non-survivors. During the first three days post cardiac surgery, impaired DO_2_ was more pronounced and in the following four days, O_2_ER was markedly lower in non-survivors, suggesting a switch from a predominantly limited oxygen supply to permanent mitochondrial dysfunction over time. (3) Our findings indicated that a reduced REE was associated with reduced 30-day, 1-year and 6-year mortality in patients after cardiac surgery.

As already reported 50 years ago by Sir Cuthbertson, we could reproduce a short ‘ebb’ phase in survivors immediately after major cardiac surgery with a negative measured energy balance compared with the calculated REE, followed by a long-lasting ‘flow’ phase characterized by increased metabolic rates compared with the predicted REE. In contrast, in non-survivors the negative measured energy balance was maintained over the entire observational period.

In the literature, a modest 7% increase in energy metabolism after uncomplicated abdominal surgery compared with the Harris–Benedict equation has been described [16]. In patients with acute pancreatitis, even a hypermetabolic state with a raised metabolic rate of up to 20–30% has been observed [17]. Moreover, a hyperdynamic cardiovascular response with an increased REE was reported in patients with uncomplicated sepsis, sepsis syndrome and septic shock. However, in line with our findings, the percent increase in the REE declined according to the sepsis severity (mean REE + 55% for uncomplicated sepsis, +24% for sepsis syndrome and +2% for septic shock) [13]. We observed a hypermetabolic state only in survivors. Non-survivors remained in a hypometabolic status throughout the observation period. In survivors, we also found a progressive increase in O_2_ER and concomitantly decreasing SvO_2_, which may have reflected a higher level of activity and a higher cellular energy demand, suggesting an improved metabolic state.

In non-survivors, the significantly reduced DO_2_ compared with survivors early after ICU admission could have resulted from a decreased intravascular volume, loss of vasomotor tone and myocardial depression, as has already been reported in sepsis patients [18,19,20]. The absent O_2_ER increase over time may have been an indication of impaired VO_2_ secondary to microcirculatory defects or deteriorated cellular respiration [3]. In line with our findings, a study of sepsis patients found an association between mortality and decreased central venous oxygen saturation (ScvO_2_) during the very first hours after ICU admission [21] whilst another study reported that maximum ScvO_2_ values during the ICU stay were associated with a higher mortality [3]. In conjunction with literature, our results suggested that, especially during the first hours of ICU admission, an impaired oxygen supply seems to be decisive in avoiding adverse outcomes by maintaining mitochondrial functions, taking into account that the Kaplan–Meier curves were separated in the early phase after surgery and proceeded in parallel thereafter.

Energy requirements greatly vary among critically ill patients, especially in those with adverse outcomes [22,23]. Generally, a measured REE defines the energy target for the prescription of nutrition [12]. However, during the early phase of an acute illness, endogenous energy production covers most of the energy needs [24]. In this phase, exogenous energy supplementation may easily exceed the energy requirements, especially when full clinical nutrition may be provided [25]. In our study, the energy requirements were reduced in non-survivors and remained low over the total observation period of seven days. However, substrate deficits due to reduced endogenous energy production or an insufficient substrate administration seem unlikely because O_2_ER was lower and, therefore, not limited by O_2_ availability; nearly all patients required insulin to maintain normoglycemia.

Our findings, therefore, highlight the importance of measuring the REE to avoid inadequate nutritional therapy, especially in vulnerable patients.

In this study, we demonstrated that a reduced REE was an independent factor for 30 day, 1-year and even 6-year mortality. Age became a more important factor for survival in the multivariate analysis only after 1 year and 6 years. A duration of ECC > 170 min was an independent factor 30 days after surgery, comprising patients with more complex surgeries such as patients receiving both a CABG and valve procedure, patients undergoing heart transplantation and patients receiving a vascular graft. Missing durations of ECC were more pronounced in patients receiving LVAD procedures.

According to our findings, the accurate determination of energy requirements in the early vulnerable phase seems to be highly relevant to optimize the metabolic demand to avoid energy imbalances. Underfeeding increases the hospital length of stay, infection rates and organ failure as well as prolonging mechanical ventilation and even increasing mortality. In contrast, overfeeding has been associated with hyperglycemia, hypertriglyceridemia, hepatic steatosis, azotemia, hypercapnia and, again, increased mortality [22,23].

As a consequence, it is important to monitor the metabolic response of patients over time either via repeated IC measurements, via the PAC or at least via the ventilator [12,26].

Our study has several limitations due to the retrospective study design. Additionally, we assessed the REE via the PAC and did not measure the REE via indirect calorimetry (IC), which is more accurate [26]. However, assessing the REE via the PAC has the advantage of being able to measure VO_2_ continuously over a prolonged period of time. Another factor introducing selection bias was that the decision to insert a PAC was based on institutional practice and the individual risk assessments of the treating physicians. Thus, only patients who were more likely to be hemodynamically unstable with a worse prognosis received a PAC and were included in this study. Hence, the mortality rate of this study cohort was higher than the mortality rate of the corresponding ICU. Further, there is no medically determined cut-off value to define a low REE; therefore, we used the median as a cut-off to divide the patients into two groups. Even though the REE is generally determined without including the body weight, we were able to rule out that a low REE was simply associated with a low BMI, size and body weight in the non-parametric group testing. Furthermore, neither BMI nor gender was a confounding factor for adverse outcomes in the uni- and multivariate cox regression models.

## 5. Conclusions

Non-survivors seemed to be unable to metabolically adapt from the early, previously-called ‘ebb’ phase to the late period, the previously-called ‘flow’ phase, subsequently remaining in a catabolic state. In those patients, our findings indicated an impaired oxygen supply early after ICU admission and persistent mitochondrial dysfunction over time. A lower REE was associated with adverse short- and long-term outcomes, emphasizing the importance of monitoring the REE in critically ill patients either via IC, the PAC or the ventilator.

## Figures and Tables

**Figure 1 nutrients-14-03366-f001:**
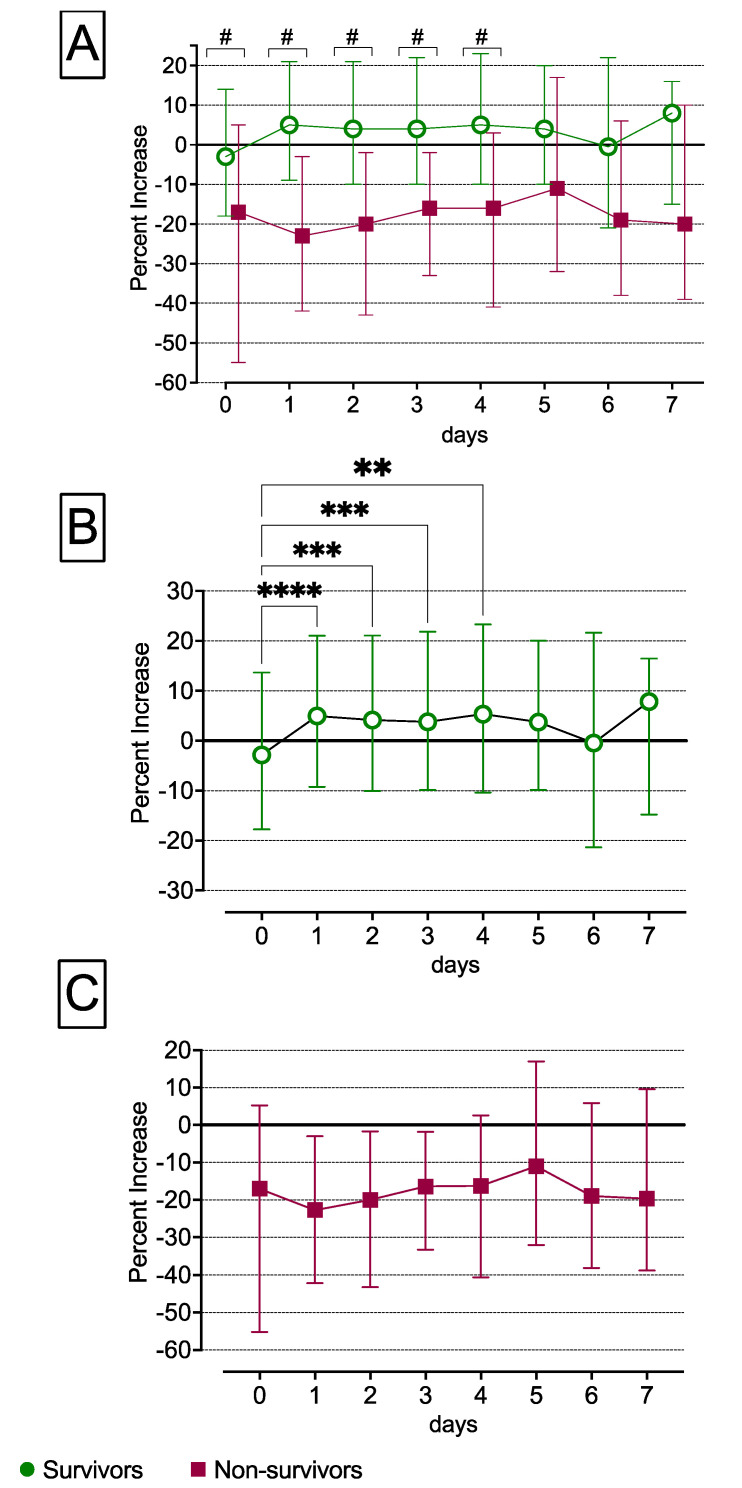
Missing transition from the early to the late period in non-survivors. In patients who survived, the REE significantly increased during the transition from the early to the late period. In contrast, the REE did not increase during the transition from the early to the late period in non-survivors, as depicted in (**A**). In survivors, there was a statistically significant percent increase in the REE from day 0 to day 1, day 2, day 3 and day 4, as depicted in (**B**). In contrast, in non-survivors, the REE did not increase during the transition from the early to the late period, as shown in (**C**). Percent increase = (REE meas − REE pred)/REE pred × 100; REE pred = 20 × kcal/kg/day; meas, measured; pred, predicted; REE, resting energy expenditure; ** *p* < 0.001, *** *p* < 0.0001, **** *p* < 0.00001; # *p* < 0.05.

**Figure 2 nutrients-14-03366-f002:**
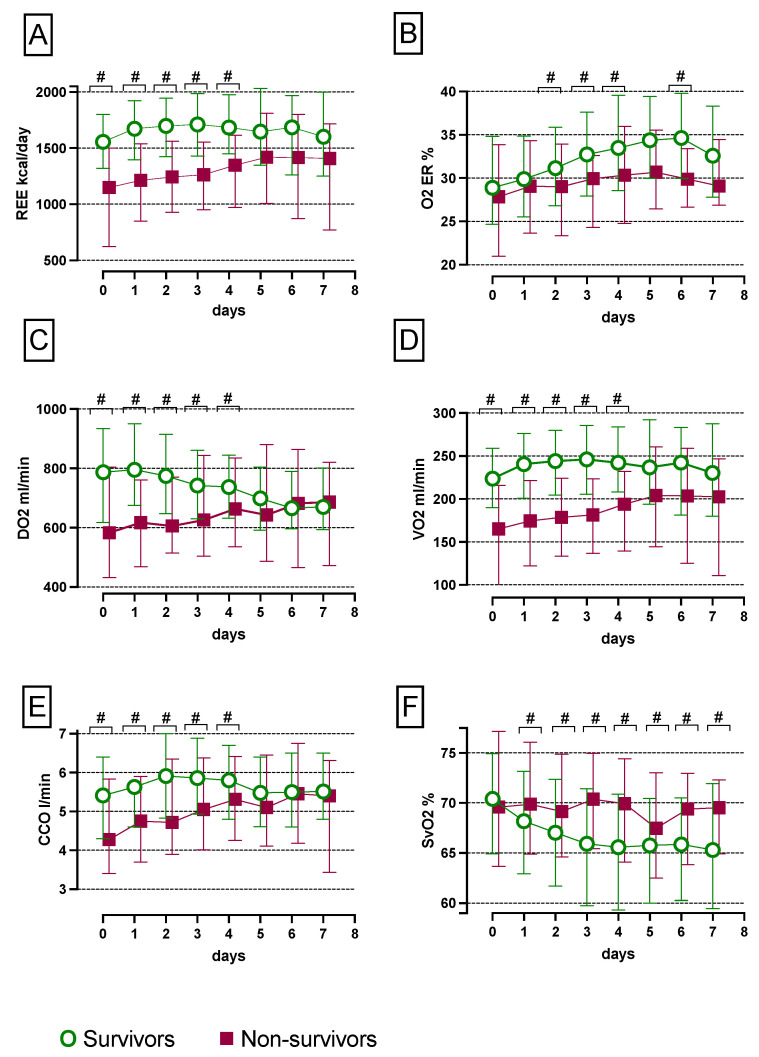
CCO, SvO_2_, DO_2_, VO_2_, O_2_ER and REE of survivors and non-survivors during the first seven days. In non-survivors, the REE median was significantly reduced compared with survivors over the first 4 days, but not on days 5, 6 and 7 (**A**). Non-survivors had significantly lower O_2_ER values on post-operative days 2, 3, 4 and 6 compared with survivors after heart surgery (**B**). DO_2_ was significantly lower in non-survivors during the first 4 days, but not on days 5, 6 and 7 (**C**). In non-survivors, VO_2_ was significantly decreased within the first 4 days (**D**). CCO was significantly lower in non-survivors over the first 5 days (**E**). SvO_2_ significantly rose from day 1 until day 7 in non-survivors compared with survivors (**F**). Patients included per day: *n* = 566; day 0: *n* = 463; day 1: *n* = 515; day 2: *n* = 405; day 3: *n* = 292; day 4: *n* = 203; day 5: 115; day 6: 73; day 7: *n* = 42. CCO, continuous cardiac output; DO_2_, oxygen delivery; O_2_ER, oxygen extraction ratio; REE, resting energy expenditure; SvO_2_, mixed venous oxygen saturation; VO_2_, oxygen consumption; # *p* < 0.05.

**Figure 3 nutrients-14-03366-f003:**
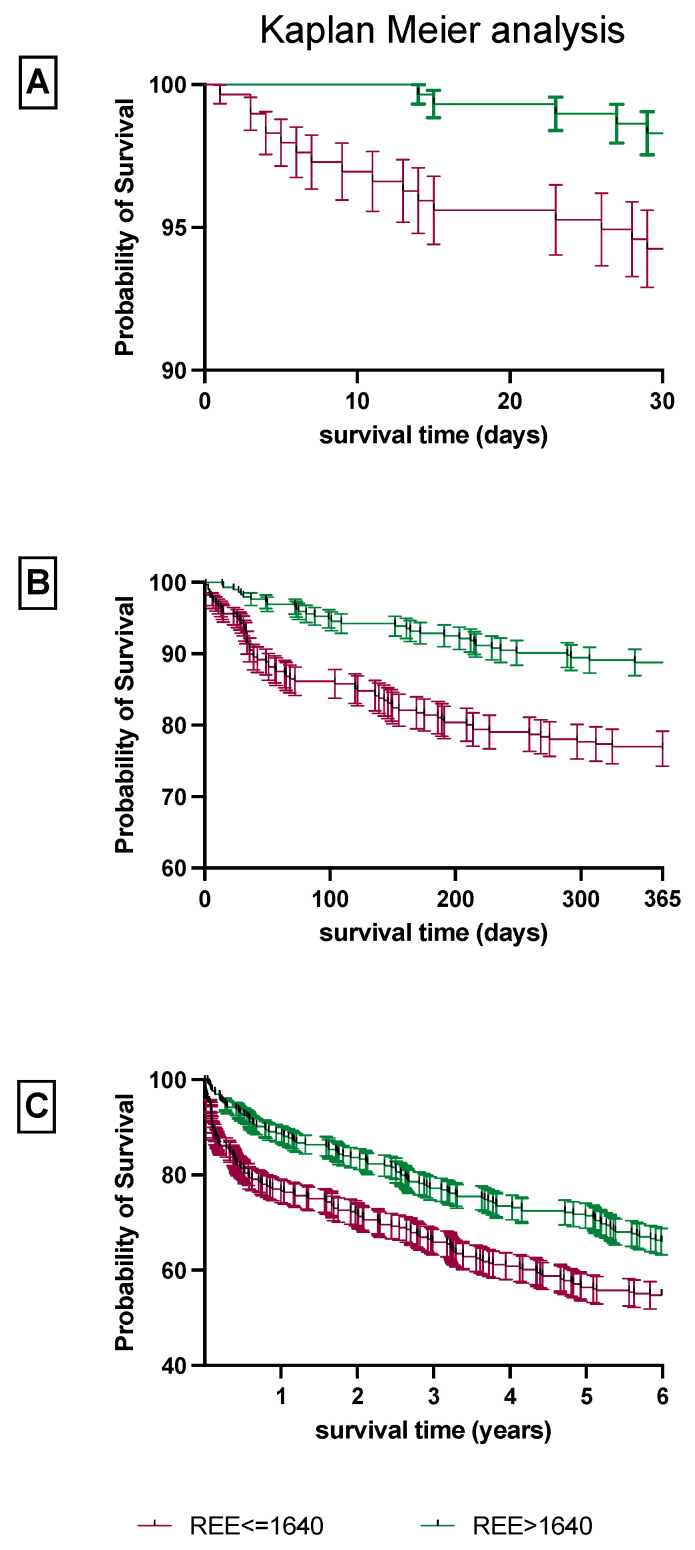
Kaplan–Meier analysis of patients with a high and low REE. Survival analysis was performed using the Kaplan–Meier for 30 days (**A**), 1 year (**B**) and 6 years (**C**). Patients were divided into 2 groups depending on their REE being either above or below the median REE of 1640 kcal/day. REE, resting energy expenditure.

**Table 1 nutrients-14-03366-t001:** The determinants of venous and arterial oxygen content, oxygen delivery, oxygen consumption, oxygen extraction and resting energy expenditure.

Determinants	
CvO_2_	=Hb × 1.37 × SvO_2_ + 0.003 × PvO_2_
CaO_2_	=Hb × 1.37 × SaO_2_ + 0.003 × PaO_2_
DO_2_	=CO × CaO_2_ × 10
VO_2_	=CO × (CaO_2_ − CvO_2_) × 10
O_2_ER	=VO_2_/DO_2_
REE meas	=VO_2_ × 4.83 kcal/L × 1.44
REE pred	=20 kcal/kg/day
Percent increase	=(REE meas − REE pred)/REE pred × 100

CvO_2_, venous oxygen content; CaO_2_, arterial oxygen content; DO_2_, oxygen delivery; meas, measured; pred, predicted; VO_2_, oxygen consumption; O_2_ER, oxygen extraction ratio; REE, resting energy expenditure.

**Table 2 nutrients-14-03366-t002:** Demographic and clinical data.

Demographic Data *n*	Total 566	Survivors 517	Non-Survivors 49	*p*-Value
**Gender**				
Male, n (%)	422 (74)	389 (75)	33 (67)	
Female, n (%)	144 (25)	128 (25)	16 (33)	0.266
**Age,** mean ± SD	63 ± 12	63 ± 12	66 ± 9	0.118
**BMI,** mean ± SD	26 ± 4	26 ± 4	26 ± 5	0.594
**Surgical procedure**				
Valve procedure, n (%)	156 (27)	148 (28)	8 (16)	
CABG, n (%)	68 (12)	64 (12)	4 (8)	
CABG and valve, n (%)	96 (17)	89 (17)	7 (14)	
Vascular graft, n (%)	15 (3)	15 (3)	0 (0)	
LVAD, n (%)	93 (16)	76 (14)	17 (34)	
HTX, n (%)	114 (20)	105 (20)	9 (18)	
Others, n (%)	24 (4)	20 (4)	4 (8)	**<0.001**
**Perioperative data**				
Lactate max, mean ± SD	3.9 ± 1.9	3.8 ± 1.7	5.2 ± 2.8	**<0.001**
HB min, mean ± SD	8.1 ± 1.3	8.2 ± 1.4	7.9 ± 1.1	0.071
PRBC count, mean ± SD	3.8 ± 3.0	3.5 ± 2.8	5.9 ± 3.9	**<0.001**
FFP count, mean ± SD	1124 ± 878	5.5 ± 4.3	6.4 ± 3.7	0.313
Blood loss mL, mean ± SD	494 ± 601	475 ± 586	786 ± 826	**0.045**
ECC min, mean ± SD	177 *±* 77	173 ± 73	232 ± 114	**0.002**
XCT min, mean ± SD	104 *±* 45	102 ± 43	131 ± 67	**0.020**

Survivors included all patients who survived hospitalization. Non-survivors comprised all patients who died at any time during hospitalization. BMI, body mass index; CABG, coronary artery bypass graft; ECC, extracorporeal circulation; FFP, fresh frozen plasma; HB, hemoglobin; HTX, heart transplantation; LVAD, left ventricular assist device; PRBC, packed red blood cells; XCT, cross-clamp time.

**Table 3 nutrients-14-03366-t003:** Univariate and multivariate cox regression analyses for REE after 30 days, 1 year and 6 years post cardiac surgery.

Cox Regression Analyses							
		Univariate Model			Multivariate Model		
		HR	CI 95%	*p*-Value	HR	CI 95%	*p*-Value
	30-day all-cause mortality						
**REE**	>1640 kcal/d **^#^**	*1.0*					
	≤1640 kcal/d	**3.5**	1.3–9.5	**0.013**	**3.2**	1.1–8.7	**0.021**
**Gender**	Male **^#^**	1.0					
	Female	1.6	0.7–4.0	0.235			
**Age**	<55 years **^#^**	1.0					
	55–65 years	1.8	0.4–7.3	0.354			
	66–75 years	2.0	0.5–7.6	0.271			
	>75 years	1.4	0.2–7.1	0.652			
**BMI**	<25 kg/m**^2^**^**#**^	1.0					
	25–30 kg/m**^2^**	0.9	0.3–2.4	0.885			
	>30 kg/m**^2^**	0.6	0.1–2.4	0.550			
	Missing	1.0	0.2–4.7	0.984			
**ECC time**	≤170 min **^#^**	1.0					
	>170 min	**3.8**	1.0–13.7	**0.040**	**3.7**	1.0–13.3	**0.043**
	Missing	**9.2**	2.4–34.7	**0.001**	**8.3**	2.2–31.5	**0.002**
**Hb min**	≥8 g/dL **^#^**	1.0					
	<8 g/dL	1.5	0.6–3.6	0.332			
	Missing	1.2	0.1–9.8	0.834			
**Lac max**	≤3.6 mmol/L **^#^**	1.0					
	>3.6 mmol/L	2.0	0.8–4.9	0.124			
**PRBCs**	≤3 units	1.0					
	>3 units	2.2	0.8–5.6	0.091			
	Missing/no PRBCs	0.5	0.1–1.9	0.334			
**FFPs**	≤4 units **^#^**	1.0					
	>4 units	3.6	0.7–17.8	0.103			
	Missing/no FFPs	1.1	0.2–4.8	0.895			
	1-year all-cause mortality						
**REE**	>1640 kcal/d **^#^**	1.0					
	≤1640 kcal/d	**2.5**	1.6–3.8	**<0.001**	**2.0**	1.3–3.1	**0.001**
**Gender**	Male **^#^**	1.0					
	Female	**1.4**	0.9–2.1	0.103			
**Age**	<55 years **^#^**	1.0					
	55–65 years	**2.5**	1.2–5.0	**0.007**	**2.8**	1.4–5.6	**0.003**
	66–75 years	**2.1**	1.1–4.3	**0.026**	**2.7**	1.3–5.4	**0.005**
	>75 years	**2.9**	1.4–6.1	**0.003**	**3.1**	1.4–6.6	**0.003**
**BMI**	<25 kg/m**^2^**^**#**^	1.0					
	25–30 kg/m**^2^**	0.7	0.4–1.1	0.207			
	>30 kg/m**^2^**	0.8	0.5–1.5	0.700			
	Missing	0.7	0.5–1.6	0.456			
**ECC time**	≤170 min **^#^**	1.0					
	>170 min	**1.7**	1.1–2.6	**0.017**	1.5	0.9–2.3	0.080
	Missing	**1.8**	1.0–3.3	**0.048**	**2.1**	1.1–4.1	**0.017**
**Hb min**	≥8 g/dL **^#^**	1.0					
	<8 g/dL	**1.7**	1.1–2.6	**0.005**	1.1	0.7–1.8	0.624
	Missing	0.2	0.03–1.9	0.201	0.2	0.0–2.0	0.017
**Lac max**	≤3.6 mmol/L **^#^**	1.0					
	>3.6 mmol/L	**1.9**	1.2–2.9	**0.002**	1.4	0.9–2.2	0.102
**PRBCs**	≤3 units	1.0					
	>3 units	**2.6**	1.6–4.0	**<0.001**	**2.4**	1.5–3.9	**<0.001**
	Missing/no PRBCs	0.5	0.2–1.0	0.063	0.7	0.3–1.3	0.298
**FFPs**	≤4 units **^#^**	1.0					
	>4 units	1.7	0.9–3.4	0.099			
	Missing/no FFPs	0.7	0.4–1.2	0.280			
	6-year all-cause mortality						
**REE**	>1640 kcal/d **^#^**	1.0					
	≤1640 kcal/d	**1.5**	1.1–2.0	**0.001**	**1.3**	1.0–1.8	**0.031**
**Gender**	Male **^#^**	1.0					
	Female	1.0	0.7–1.4	0.649			
**Age**	<55 years **^#^**						
	55–65 years	**1.7**	1.1–2.6	**0.014**	**1.9**	1.2–2.9	**0.003**
	66–75 years	**2.0**	1.3–3.1	**<0.001**	**2.4**	1.6–3.7	**<0.001**
	>75 years	**2.3**	1.5–3.7	**<0.001**	**2.5**	1.6–4.1	**<0.001**
**BMI**	<25 kg/m**^2^**^**#**^	1.0					
	25–30 kg/m**^2^**	0.7	0.5–0.9	**0.040**	0.7	0.5–1.0	0.067
	>30 kg/m**^2^**	1.0	0.7–1.5	0.687	1.3	0.9–1.9	0.113
	Missing	0.6	0.3–1.1	0.174	0.7	0.4–1.2	0.742
**ECC time**	≤170 min **^#^**	1.0					
	>170 min	1.0	0.7–1.3	0.857	**2.1**	1.5–2.8	**<0.001**
	Missing	**1.8**	1.2–2.6	0.001	0.8	0.5–1.1	0.297
**Hb min**	≥8 g/dL **^#^**	1.0					
	<8 g/dL	**1.3**	1.0–1.8	**0.014**	0.9	0.7–1.2	0.792
	Missing	0.7	0.3–1.5	0.377	0.5	0.2–1.2	0.142
**Lac max**	≤3.6 mmol/L **^#^**	1.0					
	>3.6 mmol/L	1.2	0.9–1.6	0.116			
**PRBCs**	≤3 units	1.0					
	>3 units	**1.8**	1.3–2.5	**<0.001**	**2.1**	1.5–2.8	**<0.001**
	Missing/no PRBCs	0.7	0.5–1.0	0.148	0.8	0.5–1.1	0.297
**FFPs**	≤4 units **^#^**	1.0					
	>4 units	1.5	0.9–2.6	0.087			
	Missing/no FFPs	0.7	0.6–1.6	0.784			

The univariate model was performed for demographic and perioperative characteristics. The multivariate model included only statistically significant categories of the univariate model. BMI, body mass index; CI, confidence interval; d, days; ECC, extracorporeal circulation; FFP, fresh frozen plasma; HB, hemoglobin; HR, hazard ratio; PRBC, packed red blood cells; REE, resting energy expenditure; ^#^ reference.

## Data Availability

All data generated or analyzed during this study are included in this published article.

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
