# Peer review of "Absent Metabolic Transition from the Early to the Late Period in Non-Survivors Post Cardiac Surgery"

_nutrients, 2022, doi:10.3390/nu14163366_

Round 1
Reviewer 1 Report
It is a very intersting study which showed that the relationship between REE and mortality. REE monitoring maybe used in the daily clinical setting. There are three major concerns to be mentioned.
1. The statistical analysis shoule be used repeated measure ANOVA.
2.How to treat REE should be mentioned in the discussion.
3.Can REE be compared to Pv-aCO2/Ca-vO2? Please discuss if possible.
Reviewer 2 Report
Dear Editors and Authors,
thank you for the opportunity to review this interesting paper.
This is a longitudinal analysis of the Resting Energy Expenditure (REE) and other clinical parameters in ICU patients after cardiac surgery. The parameters and their association with patient outcome (all-cause mortality) were analyzed, focusing on the changes of the measured REE during the time in comparison with the calculated REE over the course of the first 7 days after surgery.
The topic of this retrospective analysis is innovative and of interest in our field of research. The manuscript is prepared carefully, the tables and figures add to the appeal and understanding. This manuscript needs only a minor language revision, some abbreviations need introduction.
My major points are the calculation of REE and the division of patients into two groups according to the median REE, which was not normalized to body weight (in disagreement with table 1)? I am also unclear how the REE was predicted – using the same 20 kcal/kg/d for ICU days 0 - 7? On what thoughts and references were these decisions based on?
For greater details in each section, please see below.
Abstract:
Suggest to structure abstract into background, methods, results and conclusion
Please support your results by including some numbers.
Line 12: Please specify insult, as it may also be used as synonym for stroke
Line 28: please correct grammar
Lines 31-35 remain unclear to me
Introduction
Line 58: suggest to specify “in cardiac surgery”
Line 62: the same reference is cited twice
Lines 64 and 67: suggest to replace “ancient” with “former” or “previous”
Line 70: suggest to add injury severity, as well as other factors of each patient (for example….)
Methods
Line 100: The description of the inclusion criteria and procedures could be a little more precise: For example: Were elective and emergency patients included? What happened of the PAC was removed before 7 days? How often were PAC measurements recorded? In this retrospective cohort, how was the follow-up performed (consent necessary?). Was this study registered somewhere? How were survivors defined?
Table 1: I am unclear how the REE was predicted – using 20 kcal/kg/d for all ICU days, from 0 to 7? Why was this chosen?
Line 121: please correct grammar
Lines 134-135: Why was this method chosen? Why was the here REE not normalized according to body weight (in disagreement with table 1)? A REE below the chosen cutoff (since not normalized) could also simply be indicative of a lower-weight patient.
Figure 1: It would be very nice to see both graphs in one figure or at least on the same scales to be compared more easily. Were group comparisons performed? With which results? It seems this is not only about the transition, as non-survivors have a “more negative” starting point as well.
Lines 188-231: this paragraph is very hard to read and understand– suggest to put all numbers in supplementary tables and leave only the sentences in. E.g. “Non-survivors had solely significantly lower O2ER values on postoperative day 2,3,4 and day 6 compared to survivors after heart surgery [Day 2: 29 % (24, 32) vs. 31 % (25, 34); p=0.018], [Day 3: 29 % 198 (34, 32) vs. 32 % (27, 37); p=0.002], [Day 4: 30 % (24, 35) vs. 33 % (28, 39); 199 p=0.018] and [Day 6: 29 % (26, 33) vs. 34 % (29, 39); p=0.014)] as demonstrated in Figure 2B.”
Figure 2: Overall very informative, recommend adding some stars for significant differences between groups. 2A: Please indicate somewhere that this was measured REE
Line 249: I think there is a minor formatting error which removed the heading
Discussion:
Line 308: please be more precise here: “energy balance” was not topic until this line
Line 312: In which patient cohort was this described?
Line 319: how was “hypermetabolic” defined in your work?
Line 332: Please define ScvO2
Round 2
Reviewer 1 Report
I have no further question.